# Improving the Phenolic Content of Tempranillo Grapes by Sustainable Strategies in the Vineyard

**DOI:** 10.3390/plants11111393

**Published:** 2022-05-24

**Authors:** M. Esperanza Valdés, M. Inmaculada Talaverano, Daniel Moreno, David Uriarte, Luis Mancha, Mar Vilanova

**Affiliations:** 1CICYTEX-INTAEX, Technological Institute of Food and Agriculture, Av. Adolfo Suárez, s/n, 06071 Badajoz, Spain; inma.talaverano@gmail.com (M.I.T.); daniel.moreno@juntaex.es (D.M.); 2CICYTEX-FOV, Agricultural Research Institute Finca La Orden-Valdesequera, Crta. A-V, Km 372, 06187 Badajoz, Spain; david.uriarte@juntaex.es (D.U.); luisalberto.mancha@juntaex.es (L.M.); 3Instituto de Ciencias de la Vid y del Vino (ICVV), Consejo Superior de Investigaciones Científicas CSIC, Universidad de La Rioja-Gobierno de La Rioja, Finca la Grajera, Carretera de Burgos, 26080 Logroño, Spain

**Keywords:** regulated deficit irrigation, cluster thinning, phenols, skin grape, Tempranillo, semi-arid condition

## Abstract

Wine phenolics are of considerable interest due to their implication in the organoleptic appreciation of wines and due to their bioactive functions as antioxidants. In this work, the effects of sustainable strategies in the vineyard, regulated deficit irrigation treatments (RDI) and crop load level (CL) on Tempranillo grape phenolics over two seasons was studied. Rainfed (T), early (EDI) and late (LDI) regulated deficit irrigation was applied. Cluster thinning (TH) and control (C) without cluster removal were also applied under each irrigation treatment. The effect of CL remained independent of RDI for all compounds, except for phenolic acids. The RDI influence on the grape skin phenolic profile was higher than CL in the dry season (2009); however, in 2010, the effect of CL was greater. In 2009, a tendency to increase anthocyanin and hydroxycinnamic acid content in grape skins was registered in EDI with respect to T. However, significant decreases in hydroxycinnamic and flavanol compounds were found in LDI. In 2010, the wettest year, CL increased all phenolic families’ content. Thus, it can be concluded that the effects of RDI and cluster thinning treatments depend on the family of compounds considered and the meteorological conditions of the year.

## 1. Introduction

Irrigation management is a fundamental tool in order to improve grape quality, especially when the vines are cultivated in a semiarid environment. However, irrigation is not always appropriate to reach grape quality. Thus, supplying irrigation to ensure 100% potential vine evapotranspiration (ET_c_) normally reduces wine quality [1]. Nevertheless, regulated deficit irrigation (RDI) in vines has been used to improve berry and wine quality [2,3]. RDI consists of applying water in quantities smaller than those required to fully satisfy ET_c_ needs during certain periods of the growth cycle. Strategies have been designed to subject vines to greater or less severe water stress (depending on their phenological state) to achieve the desired balance between yield and quality [4]. 

Conversely, the phenological timing on vine and berry response to water stress is important [5]. In this sense, post-veraison water application was necessary to increase grape sugar level and wine alcohol content; however, water restrictions during the pre-veraison period produced more concentrated berries in terms of total phenolic and anthocyanins [6]. Girona et al., (2009) [7] also reported that water deficits applied in the early stages of the crop season had a greater effect on bud development than when applied later and thus a greater impact on the yield of the current and consecutive crop seasons. 

The overall effect of irrigation could change according to other cultural practices, particularly those affecting the crop level [8]. Cluster thinning is applied to regulate the yield levels and to help ripen the crop under poor climatic conditions or excessive crop demand [9]. In terms of fruit and wine composition, results presented in the literature have reported contrasting results. The application of cluster thinning advanced grape maturity [9] and led to better fruit quality in some cases [10], but no clear effects were observed by other authors [11,12]. Conversely, the amount of fruit removed and the timing of the operation could change the effects of this practice [13]. Mechanical cluster thinning increased the grape size in Tempranillo when the practice was performed at veraison (late thinning), but no effect was observed at pea size (early thinning), with respect to control or unthinning treatments [14]. In contrast, Fanzone et al., (2011) [15] demonstrated that cluster thinning and its timing had little to no influence on physical parameters and fruit chemical composition, and the differences with respect to the control were mainly due to the season. In addition, it has been suggested recently that the effectiveness of cluster thinning is related to the vine water regimen [15], and this may explain why the positive effect of cluster thinning often depends on the vintage [12]. There are numerous studies on the effects of water deficit and control of yield [16,17,18,19]; nevertheless, those studies were developed in others varieties and regions with other climatic conditions. Recent studies have reported that the strategy of applying moderate post-veraison water stress coupled with cluster thinning was most efficient in terms of having the potential to improve berry color and the wine of cv Tempranillo [20]. Deficit irrigation and bunch thinning improved the chemical and sensory characteristics of Tempranillo grapes and wines, showing improved color intensity, persistence and balance [21,22]. However, it is necessary to deepen the study of these techniques applied at different times and doses in a semi-arid terroir, especially regarding the phenolic composition in grape skin. 

Phenolic compounds constitute one of the most important fruit-quality parameters for winemaking, because they contribute to organoleptic characteristics such as color, astringency, and bitterness. The polyphenolic content of the berries is highly influenced by management practices in the vineyard, and their concentration and presence depend mainly on genetics, the soil, climate, and viticulture factors [23,24,25,26,27]. The information about the increase in grape quality and the changes in their profile and contents is essential for both the wine grower and wine maker. This is particularly important, as not all phenolic compounds have the same importance regarding intensity, tone and color stability of the wines [28].

The aim of the present research was to examine the effects of RDI strategies by inducing a pre- and post-veraison water deficit and crop load level on the phenolic profile composition of berries skins in cv. Tempranillo, grown under semi-arid conditions through two consecutive seasons. Our study particularly focused on profiling the different phenolic families. 

## 2. Results and Discussion 

### 2.1. Grape Skin Total Phenolic Composition

Heat, drought and light intensity are some of the environmental stress factors affecting grape development, primary and secondary metabolism, and thus the final content of polyphenols in berries [23,29]. Other studies have shown how the effects of different viticultural techniques depend on the vintage because different climatic conditions were registered [30,31]. Because the different meteorological conditions over two seasons (2009–2010) affected the phenolic content in grape skins under the treatments applied, the data analysis was performed year by year. The effects of the treatment (TR), regulated deficit irrigation treatments (RDI) and crop load level (CL), and their interaction of RDI*CL on the different families of phenolic compounds in both seasons are shown in Figure 1a,b (2009 and 2010, respectively).

Except for hydroxycinnamic compounds, no significant interaction of RDI*CL was found in the phenolic families in both years. 

The RDI treatments caused differences in the values of all phenolic families analyzed in 2009, and the same behavior was observed in 2010 with the exception of the flavonols. The CL treatments modified the concentration of the hydroxycinnamic compounds only in 2009, while in 2010, they increased the concentrations in all families of the phenolic compounds. In both years, the values of the concentrations of the different phenolic families were affected by treatments (TR), with the exception of flavanol in 2009.

Previous works have reported that the synthesis and accumulation of anthocyanin compounds in skin during berry growth can be modified by the extent and timing of water deficit [16,21,32,33,34]. In our study, dry and warm meteorological conditions during pre-veraison of 2009 caused higher water stress level vines [35]. As a result, with respect to T, deficit irrigation during the pre-veraison (EDI treatments) improved the anthocyanin concentrations in the grape skin at harvest; however, the higher water amount applied by LDI treatments during the same period decreased the levels of these phenolics. In 2010, the higher values of rainfall registered from budbreak to veraison (137 mm) caused a lower water stress in vines for this period, and lower contents of anthocyanins were found in LDI and EDI with respect to T. The CL effect showed no significant differences in 2009, but in 2010, higher anthocyanin values were found when TH was applied (*p* < 0.01). Therefore, when all treatments were compared, the highest values were achieved in EDI-TH and in T-TH in 2009 and 2010, respectively, while the lowest values were shown in LDI-C and EDI-C in the same years. 

The flavonols concentration in grape skins can be strongly affected by environmental factors, particularly sunlight exposure [36]. In 2009, the lower foliar area in T with respect to EDI and LDI (6.0, 7.2 and 7.8 m^2^/vine, respectively) should have caused a higher sunlight exposure of the clusters and could therefore favor the flavonol synthesis and accumulation in grapes skin from these vines [37,38]. However, in 2010, the wettest year, no differences were found between T and RDI treatments. The thinning treatment significantly increased the flavonol concentration in comparison to C, only in 2010 (*p* < 0.001). In this last year, a large increase was observed in LDI-TH with respect to LDI-C. Consequently, when all treatments were compared, in 2009, the highest values were observed in T-C and LDI-TH in 2009 and 2010, respectively, and the lowest in EDI-TH (2009) and EDI-C and LDI-C (2010). 

The dynamic between the hydroxycinnamic acid and its esterified forms in grape skins is unclear, and the levels tend to decline strikingly during ripening [39]. A significant interaction of RDI*CL was observed for these compounds in both seasons. Thus, when all treatments were compared, the highest values were found in EDI-C and EDI-TH in 2009 and 2010, respectively, with the lowest in LDI-C (2009) and T-C (2010).

Finally, the flavanol concentration was significantly affected by irrigation treatments in 2009 and 2010 (*p* < 0.01). The LDI (lower stress during pre-veraison) increased the flavanol concentrations, especially in 2009. Conversely, the CL showed a significant effect, increasing the flavanol values in TH with respect to C grapes in 2010 only. When all treatments were compared, no differences in inter-treatments were detected in 2009, while in 2010, the value registered in LDI-TH grape skins was higher than EDI-C. 

### 2.2. Grape Skin Individual Phenolic Composition 

Table 1, Table 2, Table 3 and Table 4 show the effects of treatment (TR), RDI and CL on individual contents of anthocyanic, flavonol, hydroxycinnamics and flavanol compounds of Tempranillo grape skin in the 2009 and 2010 seasons. 

#### 2.2.1. Anthocyanin Profile

Fifteen anthocyanic compounds were identified, quantified, and grouped into anthocyanin monoglucosides, (∑G), acetyl glucosides (∑Ac,) and coumaroyl glucosides (∑Cm) (Table 1). The results show that anthocyanidin–monoglucosides were predominant forms, followed by coumaroyl glucosides and acetyl glucoside in samples collected in both seasons. Malvidin-derivative compounds (Mv) were the predominant anthocyanin substances in grape skins of Tempranillo, regardless of the season and the treatment, while cyanidin (Cy) and petunidin (Pn) derivates exhibited the lowest values in all treatments under our experimental conditions. A similar profile was found in Tempranillo cultivar in different geographic areas of Spain [40,41].

In 2009, the Cy and Pn derivates were not affected by RDI treatments. Significant differences were found in DpG, PtG, and MvG, in their acetyl glucosides (DpA, PtA and MvA), as well as in DpC of Tempranillo grape skins from irrigation treatments EDI and LDI. In general, EDI induced the anthocyanin compound levels in grape skins with respect to grapes from nonirrigated vines (T). Thus, the highest values of monoglucosides and acetyl monoglucosides compounds were found in EDI. The effect of CL was scarce in this year. Only significant increases were found in DpA and PtA when TH was applied with respect to C. When the six combined treatments were compared, in general, the maximum values were obtained in EDI-C and EDI-TH, with the minimum obtained in LDI-C. 

The impact of RDI and CL treatments on grape phenolic composition was different in 2010. In general, in 2010, the shortened irrigation increased the concentration of individual anthocyanins. Thus, with respect to T, EDI decreased CyG and MvG contents and LDI decreased PnG and the coumaryl forms CyC and PnC. Furthermore, both deficit irrigation treatments (EDI and LDI) induced decreases in the acetyl forms DpA, PtA, PnA. In addition, ∑Ac increased when RDI was applied.

Similar to the results obtained by Girona et al., (2009) [7], cv. Tempranillo has a high phenological sensitivity to pre-veraison water stress. Thus, the application of EDI reduced pre-veraison water stress and increased anthocyanin concentrations in 2009. However, the positive effect of water stress on phenol concentration was achieved above a certain threshold [2]. In 2010, the abundant rainfall recorded during the pre-veraison period reduced the water stress of the vines in all treatments, conditioning anthocyanin biosynthesis. 

The effect of CL on anthocyanin compounds had a great impact in 2010, showing significant increases in DpG, CyG, MvG, DpA PtA, CyC and PnC. Thus, ∑G and ∑Ac were also observed when TH was applied. When all treatments were compared, significant differences were found for ∑G and ∑Ac, but not for ∑Cm. In general, the highest values were found in grapes from T-TH treatment and the lowest from LDI-C. 

Girona et al., (2009) [7] demonstrated that Tempranillo berries had great phenological sensitivity to water stress. In the same way, Koundouras et al., (2006) [42] indicated that the anthocyanin concentration in skin tissues was strongly correlated to Ψ_stem_ during early berry development, and Castellarin et al., (2007) [43] reported a higher amount of anthocyanin per berry in fruit subjected to early water deficit. In 2010, the shortened irrigation improved the concentration of individual anthocyanins. Our findings suggest that the irrigation application could cause a detrimental effect for anthocyanins synthesis when a certain level of water stress was reached. Therefore, a certain level of stress during pre- and post-veraison is suitable for the synthesis and accumulation of these compounds in this area of study. 

Franzone et al., (2011) [15] did not observe differences in the content of anthocyanin compounds of the grapes between different crop load levels. According to these authors, a significant different content of primary metabolites (mainly sugars) would be necessary to observe different amounts of the anthocyanins values. In our study, no differences in total soluble solids at harvest (in accordance with harvesting criteria, the grapes were harvested at about 23–24 °Brix) with a foliar area-to-yield ratio between TH and C vines (2.0 and 2.4 m^2^/kg, respectively), were found in 2009. Thus, there was not an apparent sugar limitation, and thinning did not affect the biosynthesis of anthocyanins. However, in 2010, the foliar area-to-yield ratio was significantly modified in TH with respect to C (1.1 vs. 1.5 m^2^/kg), thus affecting the source–sink balance, which has been proven to be an effective tool in modulating sugar accumulation at harvest in berries and is related with anthocyanin gene expression [43,44,45,46]. Hence, the negative impact caused by water status generated in anthocyanin values in rainy years could be improved with the implementation of cluster thinning in the vineyard. Therefore, our results highlight the great role that cluster thinning plays on the different water statuses and their effects on grape quality.

#### 2.2.2. Flavonol Profile

As Table 2 shows, myricetin (My) followed by quercetin (Qc) compounds were the most abundant flavonols on the profile of cv. Tempranillo skins in both seasons. This profile was similar to what was previously reported for this cultivar [41,46]. However, the quercetin flavonols were predominant in other studies of Spanish (Tempranillo, Garnacha and Garnacha Tintorera) and French varieties (Cabernet Sauvignon, Merlot, Syrah, and Petit Verdot) [47]. 

With the exception of My, all flavonol compounds in 2009 were affected by RDI treatments. Thus, EDI and LDI treatments led to lower contents of quercetine (Qc) and isorhamnetin (Ih) values with respect to T. Similarly, kaempferol (Kp) substances reached lower values than T in EDI treatments. Myricetin (My) was affected by CL in that its concentration decreased when TH was applied with respect to control. The interaction of RDI*CL was only significant for Ih in this season. In the 2010 season, EDI induced a decrease in My and Ih vs. T. In addition, CL had a positive effect, increasing all groups of compounds when TH was applied, especially in the vines subjected to LDI treatment. In general, the highest values of compounds determined were reached in TH and LDI-TH for 2009 and 2010, respectively. 

It has been reported that sun exposure of the berries has a considerable effect on flavonol content [48,49,50]. According to Fanzone et al. (2011) [15], higher solar radiation can induce a greater expression and regulation of flavonol synthesis in berries. The application of cluster thinning directly after the fruit sets allowed for a greater availability of photoassimilates in the leaves at early stages of berry growth [15].

In 2009, a significant increase in leaf area was observed in irrigated vines with respect to rainfed vines (T = 6.0; EDI = 7.2 and LDI = 7.9 m^2^/vine). These results suggest that the accumulation of flavonol compounds did not depend strictly on water stress timing, but in indirect ways, through differences in the leaf area. Regarding the effect of CL, previous studies have suggested certain developmental or sugar regulation of the flavonol pathway [44,51]. When TH is applied at early stages, it allows for greater availability of photoassimilates (sugars) in leaves. Thus, the higher rate of flavonols might be explained by an inductive effect of sugars on the expression of genes involved in biosynthesis [51]. 

Recent studies have shown that in the absence of UV-B radiation, flavonol biosynthesis was virtually undetectable until veraison, when a rapid increase in the concentration of these substances in the skin was observed, suggesting that sugars may be involved in the flavonol biosynthetic pathway [51]. Therefore, the higher concentrations of flavonols in the TH-skins could be explained on the basis that, at the time of biosynthesis, a higher concentration of sugars in these berries would have induced a higher expression of genes involved in the biosynthesis of these compounds. Conversely, other authors [15] have also shown that the effects of thinning on flavonol concentration depend on solar radiation and temperature. In our study, higher differences in flavonol concentration between C and TH were shown in 2010 versus in 2009, because higher temperatures were reached in the 2010 season. 

#### 2.2.3. Flavanols 

Table 3 shows the effects of RDI and CL treatments on the flavanol profile of Tempranillo skins. The highest concentrations were recorded by PB3 (accounting approximately between 53–67% and 43–55%, of the total flavan-3-ols, in 2009 and 2010, with respectively, followed by PB1 (25–38% in 2009, and 39–51% in 2010) and (+)-catechin (27–34% in 2009, and 20–28% in 2010). Finally, (-)-epicatechin was a minor contributor to total flavanol compounds. 

In 2009, a significant increase in all flavanol compounds determined in the skin were registered in the EDI and LDI treatments compared with T, with larger increases in LDI than in EDI. In 2010, no significant differences were found. Therefore, the water stress was not beneficial for the synthesis and accumulation of flavonols; the amount of water applied during pre-veraison had more effect than during post-veraison. Cluster thinning increased the concentrations of EPI and PB1compounds in 2009, and only PB1 was affected in 2010, with an increased concentration. When all treatments were taken into account, LDI-TH and T-C skins showed maximum and the minimum values, respectively, as the general trend in both years.

Many studies have reported that the accumulation of flavan-3-ol in skin grape occurs during veraison, while post-veraison, the level was relatively constant [52,53]. Pre- and post-veraison application of water deficit increased proanthocyanindin levels in Syrah and Cabernet Sauvignon berries, but only transiently, and at harvest, no differences were observed [52,53]. Thus, because contrasting results among studies have been reported, the effect of irrigation on these substances still remains unclear [54,55]. 

The application of LDI reduced water stress in the grapevine along the same period, and it could have increased the flavan-3-ol concentration. Conversely, some authors have reported that the skins from well-ripe grapes release higher amounts of procyanidin than those from unripe grapes [53,56]. Because the general trend of most of the compounds was an increase in TH skins, our results suggest that TH grapes reached a higher level of phenolic maturity than C. The increase registered in the concentration of flavanols in the grape skin from thinned vines vs. non-thinned ones in veraison was observed in other studies carried out in cv. Syrah [10]. Another study analyzed the effects of thinning at different times for Malbec, where only increases in the concentration of flavonols was observed at fruit set [15]. Increases in the concentration of flavanols were also found in wines from thinned treatments carried out when the berry reached pea size in cv. Merlot and Cabernet Sauvignon [57] and when thinning was performed at the beginning of veraison in cv. Tempranillo and Garnacha [58].

The levels of flavan-3-ol, especially procyanidin, together with the anthocyanin content, represent the ability to obtain wines with the potential for aging, because these compounds are related to color stability and body, through reactions of copigmentation and polymerization [59]. Caution should be taken since the same compounds, in excess, could result in higher astringency and bitterness [60], which is particularly important for young wines. 

#### 2.2.4. Hydroxycinnamic Compounds and Stilbene

Table 4 shows the influence of RDI and CL on grape-skin hydroxycinnamic acids, their derivates, and on stilbene concentration. All compounds were affected by RDI treatment in both seasons; however, only CF-T and COU-T, as well as CF, CF-T and COU-T were affected by CL treatment in 2009 and 2010, respectively. Coumaryl compounds (COU and COU-T) were the most abundant compounds in the skin of cv. Tempranillo. With respect to the effects of RDI treatments, different trends were observed depending on the deficit irrigation period, year and compound affected. Early deficit irrigation (EDI) treatment increased the contents of COU-T with respect to T in both years; however, there was a decrease in 2010 when LDI was applied. Moreover, CF-T concentration showed an increase when EDI was applied in 2010, and COU also increased in both RDI treatments (2010).

Conversely, in the 2009 season, thinning increased CF-T and decreased COU-T content with respect to C. In 2010, the TH berries showed higher contents of CF-T, COU-T and CF compared with C berries. When all treatments were observed, a significant increase was reached by CF, COU, FE, and COU-T when EDI-TH was applied in the 2010 season. Significance was also found in RDI*CL for COU-T in both seasons of the study. 

A few studies had focused on the effect of RDI in hydroxycinnamic compounds. Niculcea et al. [61] reported that CF-T, the major hydroxycinnamic compound in that work, increased when deficit irrigation was applied with respect to well-watered Tempranillo vines. Another study also reported that cluster thinning increased the hydroxycinnamic concentration with respect to control [58]. 

The RDI treatment significantly affected the Trans-R concentration values, but this compound was not affected by CL (Table 5). In both seasons, EDI increased its contents vs. T. A significant reduction in resveratrol content in response to water stress in Shiraz and Cabernet Sauvignon at harvest was observed [62]. In addition, Deluc et al., (2008) reported that the water deficit in pre-veraison increased stilbene metabolism in Cabernet Sauvignon berries with respect to the well-watered ones. The water deficit also increased the accumulation of trans-piceid (the glycosylated form of resveratrol) by five-fold in Cabernet Sauvignon berries but not in Chardonnay [63].

#### 2.2.5. Principal Component Analysis (PCA)

Principal component analysis was performed on the phenolic compound families of Tempranillo grape skins. The first PCA (Figure 2a) was performed on data from the 2009 season. The two principal components, PC1 and PC2, accounted for 89.15% of the total variance (57.82% and 31.34%, respectively). PC1 was strongly positively correlated with flavanol compounds and negatively correlated with anthocyanin and hydroxycinnamic compounds. The flavanol family was positively correlated with PC2. A clear effect from the RDI treatments was shown. However, no discrimination was found for TH treatments on the phenolic composition of the skins. Thus, T-C, T-H, EDI-C and EDI-TH were situated on the negative side of PC1, with LDI-C and LDI-TH on positive side of the same axis. According to the figure, three groups were formed, a first group, EDI (C and TH), was correlated with the highest concentration of anthocyanin and hydroxycinnamic acids. A second group, T (C and TH), was correlated with high flavanol and the last group, LDI, with flavonol content. The second PCA (Figure 2b) was performed with the data from the 2010 season. The PCA accounted for 91.47% of the total variance (64.26% and 27.21%, in PC1 and PC2, respectively). A good separation among treatments was observed, where CL was more discriminant than RDI. Thus, TH treatments were situated on the positive side of PC1, whereas C was located on the negative side. PC1 was defined by anthocyanin and flavonol, and flavanols and PC2 by hydroxycinnamic compounds, both on the positive side of these axes. T-TH and LDI-TH showed high values of flavonol and anthocyanin compounds, while EDI-TH exhibited the highest values of hydroxycinnamic acids.

## *3.* Materials and Methods

### 3.1. Plant Material

The experiment was carried out at the experimental vineyard cv. Tempranillo from the Finca La Orden-Valdesequera Agricultural Research Centre (Regional Government of Extremadura) in Badajoz, south-western Spain (lat: 38°51′ N, long: 60°4′ W; alt: 198 m) over two consecutive seasons (2009–2010). The vineyard, which was planted in 2001 (2.5 × 1.2 m with 3333 vines/ha), was oriented north-west to south-east. The vines are grafted onto 110 Richter rootstock. A double Royat cordon pruning system is followed and the shoots are vertically positioned. Six spurs per vine and two buds per spur were retained during winter pruning. The soil at the site has a silt-loam texture.

### 3.2. Experimental Design

The experiment had a randomized block design with three treatments replicated across four blocks. The treatments were rainfed (T), and two RDI treatments established according to the crop evapotranspiration (ET_c_): early RDI (EDI), providing 25% of ET_c_ during pre-veraison and 75% of ET_c_ post-veraison; and late RDI (LDI), providing 75% of the ET_c_ during pre-veraison and 25% post-veraison. In 2010, due to the different meteorological conditions, the provision of ET_c_ was 19% instead of 25%, and 56% instead of 75% in post-veraison (EDI). 

Irrigation for EDI started when midday stem water potential (Ψ_stem_) values reached −1.0 MPa [36], and in LDI, it started when Ψ_stem_ values reached −0.6 MPa [64]. After veraison induced a moderate water deficit [29] in EDI, the irrigation was withheld until Ψ_stem_ value reached −0.8 MPa. In LDI, the irrigation was withheld until Ψ_stem_ values reached −1.2 MPa. All irrigation water was provided using pressure-compensated emitters (4 L/h) located between rows, 60 cm apart.

The water consumption of grapevines (ET_c_) was calculated from the weight differences recorded in the lysimeter in the vineyard between two consecutive measurements. The lysimeter was located in the experimental vineyard with two vines under non-water deficit to reproduce the cultivation conditions. More details about the lysimeter data collection and vine water use are reported in [64]. The reference evapotranspiration (ET_o_) value was calculated according to the Penman–Monteith method [65]. The crop coefficient K_c_ was calculated on a daily basis as the ratio of ET_c_/ET_o_. The crop load was fixed in winter pruning to 4 budsm^2^ for all treatments. 

After fruit set, two-load levels for each water status were established: control treatment (C) and cluster thinning (TH) treatment, in which the load was adjusted to 4 cluster/m^2^ of planting area by removing clusters after fruit set. 

In summary, the treatments were two levels of crop load (C and TH) in the three levels of water status (T, LDI and EDI) as follows: T-C, T-TH, LDI-C, LDI-TH, EDI-C and EDI-TH. 

In accordance with the harvesting criteria, the grapes were harvested with similar soluble solids (about 23–24 °Brix). In this sense, vines from the TH treatment were harvested nine and twelve days earlier than C for the 2009 and 2010 seasons, respectively.

### 3.3. Climate Conditions

The climate conditions of the area are defined as hot according to the Geoviticulture MCC classification system [66]. Meteorological data were obtained during the experiment from an agro-climate station (Network of Extremadura Advice to Irrigation; REDAREX) located 100 m from the vineyard. The treatment EDI provided the most irrigation water in both years, 2009 and 2010, with a respective additional 58 and 78 mm compared with LDI (Table 5).

### 3.4. Leaf Area

Leaf area index (LAI) was determined during the growing season with an LAI-2000 plant canopy analyzer (LI-COR Inc., Lincoln, NE, USA) for two vines per experimental plot (eight vines per treatment). Measurements were carried out directly before dawn under diffuse radiation using a 270° view cap (LI-COR LAI-2000 Manual). An initial reference reading was taken above the canopy, and eight readings were taken below the canopy to include the entire soil area allotted per vine [22].

### 3.5. Phenolic Compounds Analysis 

The extraction of phenolic compounds in grape skin was applied on 50 berries from each experimental plot in triplicate. The extraction was developed according to a previous work [32]. The skin phenols were analyzed by HPLC Agilent Model 1200 LC instrument (Agilent Technologies, Palo Alto, CA, USA), equipped with a degasser, quaternary pump, column oven, autosampler Agilent 1290 infinity, UV–Vis diode-array detector (DAD), and fluorescence spectrophotometer detector (FLD). Separation was performed on a column, Ace^®^ 5 C18 250 × 4.6 mm (Advanced Chromatography Technologies, Aberdeen, Scotland), used as the stationary phase. Chemstation software package was used to control the instrument and for data acquisition and data analysis. 

The analysis was carried out according to Gómez Alonso [67] with some modifications [32]. Phenolic compounds were identified according to the retention times of the commercial standards and the UV–Vis data obtained from pure compounds and/or published in previous studies [67]. Quantification of non-commercial compounds was made using the straight calibration compound belonging to the same family and following the order of elution. Hence, malvidin-3-O-glucoside was used for anthocyanins, myricetin-3-O-glucoside for flavonols, catechin for flavanols, caffeic acid for hydroxycinnamic acids and trans-resveratrol for stilbenes. Concentrations in grape samples were expressed as milligrams per weight of fresh weight (mg/kg berry fresh).

The anthocyanins present in extracts were identified in the monoglucoside (G) forms of delphinidin (DpG), cyanidin (CyG), petunidin (PtG), peonidin (PnG), and malvidin (MvG); in the acetylglucoside (A) forms (DpA, CyA, PtA, PnA and MvA) and in the p-coumaroylglucoside (C) forms (DpC, CyC, PtC, PnC and MvC). Sums of the nonacylated, acetyl acylated, and coumaroyl acylated anthocyanins were expressed in mg/kg berry fresh of malvidin-3-glucoside. The flavanols analyzed were (+)-catechin (CAT), (-)-Epicatechin (EPI), and procyanidins (B1, B2 and B3). The flavonols analyzed were classified by their aglycone backbone: myricetin-type (My) (myricetin 3-glucoside); quercetin-type (Qc) (quercetin 3-glucoside, quercetin 3-rutinoside, quercetin 3-glucuronide and quercetin 3-galactosie); kaempherol-type (Kp) (kaempferol 3-glucoside and kaempherol 3-rutinoside) and isorharmnetin-type (Ih) (isorhamnetin 3-glucoside). Non flavonoid compounds analyzed were hydroxycinnamic acids (caffeic (CF), coumaric (COU), and ferulic (FE)), and hydroxycinnamyl-tartaric derivates, caftaric (CF-T), coutaric (COU-T), and fertaric (FE-T), and stilbenes (t-resveratrol (trans-R)).

### 3.6. Statistical Analysis

All data were analyzed for statistical significance by two-way ANOVA: regulated deficit irrigation (RDI), crop load level (CL), and the interaction of RDI*CL. To test significant differences among the six treatments (TR), the data were submitted to ANOVA. A comparison of means was performed using Tukey’s multiple range test at *p* < 0.05. Principal component analysis (PCA) was performed to graphically visualize discrimination among treatments on the basis of the polyphenolic family concentrations. The data were analyzed by using XLstat-Pro (2011 Version, Addinsoft, Paris, France).

## 4. Conclusions

Among the many bioactive compounds contained in wine, the polyphenols represent the main class, mainly in red wines. The content and profile of polyphenols in grape is affected by the cultivar and environmental factors. In this work, the effect of regulated deficit irrigation (RDI) and crop load level (CL) on grape skin phenolic compounds of cv Tempranillo depended on the season and on the family of compounds considered. In the dry season of 2009, a higher influence was observed in RDI application with respect to CL on phenolic compounds, increasing the concentration of anthocyanin and flavanol compounds. However, when the water status was not a limiting factor (2010 season), TH induced an increase in most families of compounds analyzed. A positive effect of RDI combined with cluster thinning (TH) treatment was observed on the anthocyanin compounds during the dry season of 2009, when RDI was applied pre-veraison (EDI). In addition, EDI-TH increased the concentration of hydroxycinnamic acids when water was available in the vineyard during the growing season. These results suggest that there is a close link between regulated deficit irrigation (RDI) and crop load level (CL) treatments and grape phenolic composition. Both treatments (RDI and CL) could be good tools to modulate phenolic grape composition and therefore improve wine quality.

## Figures and Tables

**Figure 1 plants-11-01393-f001:**
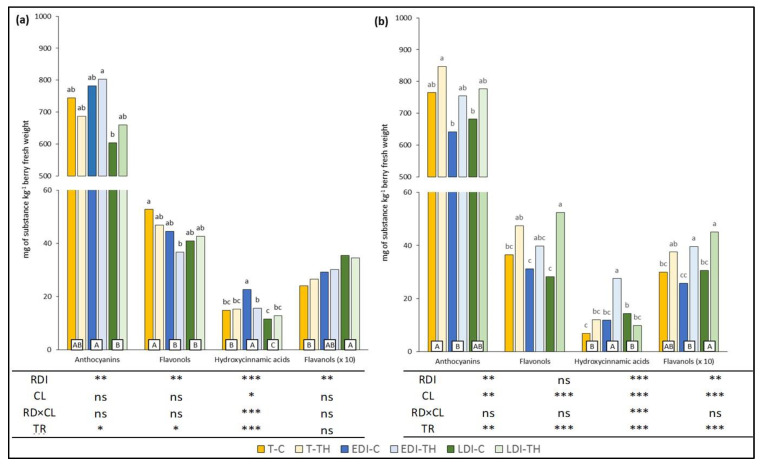
Effects of regulated deficit irrigation (RDI) and crop load level (CL) on total anthocyanins, flavonols, phenolic acids and flavanol content in 2009 (**a**) and 2010 (**b**). RDI, regulated deficit irrigation; CL, cluster loading level; TR, treatment. T, rainfed, EDI and LDI, regulated deficit irrigation in pre- and post-veraison, respectively. C, control; TH, cluster thinning. Different lower case letters are used to indicate significant differences for treatment factor and different capital letters for irrigation factor (*p* < 0.05). *, **, *** and ns indicate significant differences at *p* < 0.05, *p* < 0.01, *p* < 0.001 and no significance, respectively.

**Figure 2 plants-11-01393-f002:**
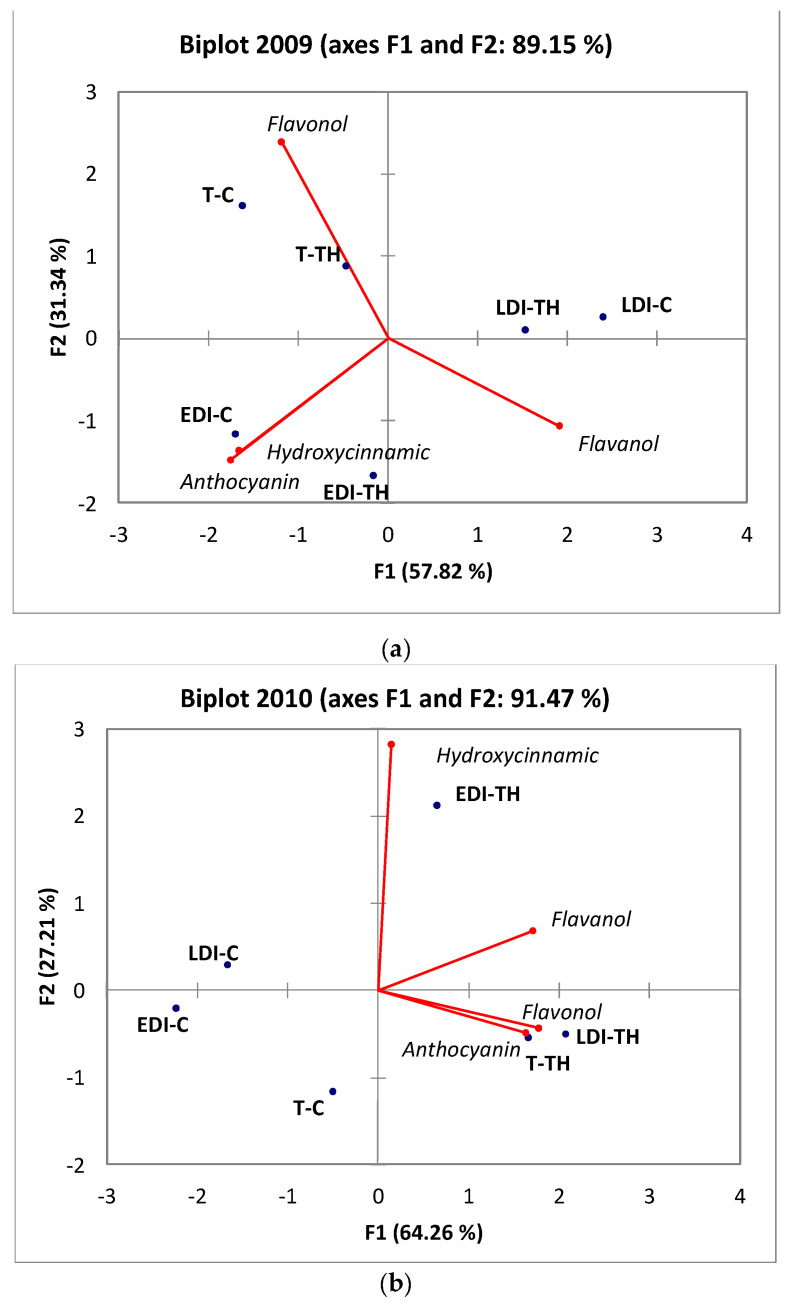
Principal component analysis results: (**a**) 2009 season; (**b**) 2010 season. T, rainfed; EDI and LDI, regulated deficit irrigation in pre- and post-veraison, respectively; C, control; TH, cluster thinning; RDI, regulated deficit irrigation; CL, crop load; C, control; TH, cluster thinning.

**Table 1 plants-11-01393-t001:** Effects of regulated deficit irrigation and cluster thinning on grape skin anthocyanin concentration (mg/kg berry fresh weight) in 2009 and 2010.

	DpG	CyG	PtG	PnG	MvG	∑G	DpA	CyA	PtA	PnA	MvA	∑Ac	DpC	CyC	PtC	PnC	MvC	∑Cm
2009																		
T	85.5 ab	9.9	85.4 ab	18.4	266.4 ab	476.9 ab	4.8 a	1.1	6.5 a	0.9	26.6 a	40.5 a	3.2 a	3.8	39.0	7.4	142.8	201.1
EDI	96.4 a	13.4	92.4 a	22.8	304.4 a	542.1 a	4.8 a	1.2	6.2 a	0.9	27.0 a	40.8 a	2.9 ab	4.0	38.2	7.6	152.8	210.5
LDI	75.3 b	11.2	71.7 b	18.3	230.7 b	417.0 b	3.8 b	1.0	4.8 b	0.7	22.0 b	32.9 b	1.6 b	3.5	33.1	6.8	132.8	182.2
C	83.4	11.2	81.4	20.1	268.4	475.6	4.1 b	1.0	5.4 b	0.8	24.3	36.2	2.6	3.5	35.8	7.2	145.3	199.2
TH	88.1	11.8	85.0	19.6	266.0	481.8	4.8 a	1.2	6.3 a	0.9	26.1	39.9	2.6	4.0	37.7	7.4	140.2	196.7
T-C	87.1	11.5	87.1 ab	21.0	273.7 ab	492.0 ab	4.6 ab	1.1	6.3 ab	1.0	27.4	41.1 ab	3.9	3.8	40.4	7.7	151.4 ab	212.2
T-TH	83.8	8.3	83.7 ab	15.9	259.2 ab	461.8 ab	5.0 a	1.1	6.6 a	0.8	25.8	39.9 ab	2.6	3.9	37.6	7.1	134.3 ab	190.1
EDI-C	98.4	13.3	92.6 a	23.3	308.7 a	549.2 a	4.6 ab	1.1	5.6 ab	0.8	24.7	37.3 ab	2.0	3.7	35.3	7.6	143.3 ab	196.6
EDI-TH	94.4	13.5	92.2 a	22.2	300.0 a	535.1 a	5.1 a	1.2	6.8 a	1.1	29.4	44.4 a	3.7	4.3	41.0	7.6	162.3 a	224.3
LDI-C	64.6	8.8	64.3 b	15.9	222.7 b	385.6 b	3.0 b	0.9	4.2 b	0.5	21.0	30.2 b	1.8	3.1	31.7	6.2	141.4 ab	188.8
LDI-TH	86.0	13.5	79.1 ab	20.7	238.7 ab	448.5 ab	4.5 ab	1.2	5.4 ab	1.0	23.0	35.6 ab	1.4	3.9	34.5	7.4	124.2 b	175.7
Significance																		
RDI	*	ns	*	ns	**	**	*	ns	**	ns	*	*	*	ns	ns	ns	ns	ns
CL	ns	ns	ns	ns	ns	ns	*	ns	*	ns	ns	ns	ns	ns	ns	ns	ns	ns
RDI*CL	ns	ns	ns	ns	ns	ns	ns	ns	ns	ns	ns	ns	ns	ns	ns	ns	ns	ns
Treatment	ns	ns	*	ns	*	*	*	ns	*	ns	ns	*	ns	ns	ns	ns	*	ns
	Dp	Cy	Pt	Pn	Mv	∑G	DpA	CyA	PtA	PnA	MvA	∑Ac	DpC	CyC	PtC	PnC	MvC	∑Cm
2010																		
T	69.9	8.3 a	70.8	27.0 a	273.7 a	460.5 a	5.1 a	1.1	7.6 a	1.7 a	38.8	55.1 a	1.2	4.5 a	49.2	9.5 a	219.4	290.9
EDI	52.8	4.5 b	61.1	19.0 ab	233.9 b	380.3 b	4.3 b	1.0	6.7 b	1.0 b	35.9	49.7 b	1.2	3.7 ab	45.1	8.2 ab	203.5	268.2
LDI	58.6	6.8 ab	65.7	17.0 b	251.7 ab	409.6 ab	4.3 b	1.0	6.5 b	0.9 b	35.7	49.2 b	1.0	3.5 b	45.6	7.7 b	206.2	270.4
C	49.8 b	5.0 b	53.0	19.1	235.3 b	371.0 b	4.0 b	0.9	6.6 b	1.1	36.1	49.6 b	1.1	3.4 b	44.9	7.8 b	211.9	275.7
TH	71.0 a	8.1 a	78.8	22.9	270.9 a	462.6 a	5.1 a	1.2	7.2 a	1.3	37.5	53.1 a	1.2	4.4 a	48.3	9.1 a	207.5	277.3
T-C	57.6	6.0 bc	59.4 ab	25.1	259.2 ab	417.1 abc	4.5 ab	0.9	7.3 a	1.7 a	38.7	54.0 ab	1.1	4.3 ab	48.7	9.4 a	223.4	294.0
T-TH	82.2	10.5 a	82.2 a	28.9	288.2 a	503.8 a	5.7 a	1.3	7.8 a	1.7 a	38.8	56.2 a	1.3	4.8 a	49.7	9.6 a	215.5	287.7
EDI-C	43.7	3.9 c	48.6 b	16.3	207.2 b	327.5 c	3.7 b	0.9	6.5 ab	0.8 c	35.1	47.8 ab	1.1	3.0 b	43.2	7.1 b	205.9	266.6
EDI-TH	61.8	5.2 bc	73.5 ab	21.7	260.6 ab	433.1 abc	4.9 ab	1.1	6.9 ab	1.2 abc	36.7	51.6 ab	1.2	4.5 a	47.1	9.4 a	201.1	269.8
LDI-C	48.1	5.1 bc	50.9 b	16.0	239.4 ab	368.3 bc	3.9 b	0.9	6.0 b	0.8 bc	34.4	46.9 b	1.0	2.9 b	42.9	6.9 b	206.4	266.5
LDI-TH	69.2	8.5 ab	80.6 ab	18.0	263.9 ab	450.8 ab	4.7 ab	1.1	6.9 ab	1.0 abc	37.1	51.6 ab	1.0	4.0 ab	48.3	8.4 ab	206.0	274.3
Significance																		
RDI	ns	***	ns	*	*	*	*	ns	**	**	ns	*	ns	**	ns	*	ns	ns
CL	*	***	***	ns	**	***	**	ns	*	ns	ns	*	ns	**	ns	*	ns	ns
RDI*CL	ns	ns	ns	ns	ns	ns	ns	ns	ns	ns	ns	ns	ns	ns	ns	ns	ns	ns
Treatment	ns	***	***	ns	*	**	**	ns	**	**	ns	*	ns	**	ns	*	ns	ns

DpG, delphinidin-3-glucoside; DpA, delphinidin-3-glucoside acetate; DpC, delphinidin-3-glucoside coumarate; CyG, cyanidin-3-glucoside; CyA, cyanidin-3-glucoside acetate; CyC cyanidin-3-glucoside coumarate; PtG, petunidin-3-glucoside; PtA, petunidin-3-glucoside acetate; PtC, petunidin-3-glucoside coumarate; PnG, peonidin-3-glucoside; PnA, peonidin-3-glucoside acetate; PnC, peonidin-3-glucoside coumarate; MvG, malvidin-3-glucoside; MvA, malvidin-3-glucoside acetate; MvC, malvidin-3-glucoside coumarate; ∑G, global monoglucosides forms; ∑Ac, global acetylglucoside forms; ∑Cm, global coumaroylglucoside forms, RDI, regulated deficit irrigation; T, rainfed; EDI and LDI, regulated deficit irrigation; CL, crop load; C, control; TH, cluster thinning, Within each column, different letters indicate significant difference among treatments after Tukey test at *p* < 0.05, *, **, *** and ns indicate significant differences at *p* < 0.05, *p* < 0.01, *p* < 0.001 and no significance respectively.

**Table 2 plants-11-01393-t002:** Effects of regulated deficit irrigation and cluster thinning on grape skin flavonol concentration (mg/kg berry fresh weight) in 2009 and 2010.

	My	Qc	Kp	Ih
2009				
T	30.3	10.3 a	2.6 a	4.6 a
EDI	24.7	8.6 b	1.6 b	3.8 b
LDI	26.3	8.3 b	1.9 ab	3.3 b
C	29.2 a	9.0	1.8	4.0
TH	25.0 b	9.2	2.2	3.8
T-C	33.7	9.7 ab	2.1 ab	4.8 a
T-TH	26.9	11.0 a	3.0 ab	4.3 ab
EDI-C	27.7	9.2 ab	1.5 b	4.2 ab
EDI-TH	21.8	8.0 b	1.7 ab	3.5 bc
LDI-C	26.2	8.1 b	1.7 ab	3.1 c
LDI-TH	26.4	8.6 ab	2.0 ab	3.6 bc
Significance				
RDI	ns	**	*	***
CL	*	ns	ns	ns
RDI*Cl	ns	ns	ns	*
Treatment	ns	*	*	**
2010				
T	24.4 a	9.4	1.3	4.9 a
EDI	19.7 b	8.7	1.2	4.3 b
LDI	22.8 ab	9.3	1.6	4.8 ab
C	17.6 b	7.5 b	1.2 b	4.2 b
TH	26.9 a	10.8 a	1.6 a	5.1 a
T-C	19.8 b	9.2	1.2 ab	4.6 ab
T-TH	28.9 a	9.6	1.4 ab	5.2 a
EDI-C	16.6 b	7.9	1.2 ab	4.0 b
EDI-TH	22.8 ab	9.4	1.2 ab	4.6 ab
LDI-C	16.5 b	5.3	1.1 b	4.1 b
LDI-TH	29.1 a	13.4	2.1 a	5.5 a
Significance				
RDI	*	ns	ns	*
CL	***	*	*	***
RDI*CL	ns	ns	ns	ns
Treatment	***	ns	*	***

My, myricetine compounds; Qc, quercetine compounds; Kp, kaempherol compounds; Ih, isorhamnetine compounds. RDI, regulated deficit irrigation, T, rainfed; EDI and LDI, regulated deficit irrigation; CL, crop load; C, control; TH, cluster thinning. Within each column, different letters indicate significant difference among treatments after Tukey test at *p* < 0.05. *, **, *** and ns indicate significant differences at *p* < 0.05, *p* < 0.01, *p* < 0.001 and no significance, respectively.

**Table 3 plants-11-01393-t003:** Effects of regulated deficit irrigation and cluster thinning on grape skin flavanol concentration (mg/kg berry fresh weight) in 2009 and 2010.

	CAT	EPI	PB1	PB2	PB3
2009					
T	1.43 b	0.03 b	1.37 c	0.44 b	2.96 b
EDI	1.87 ab	0.04 b	2.11 b	0.60 ab	3.53 ab
LDI	2.26 a	0.26 a	2.63 a	0.80 a	3.95 a
C	1.74	0.06 b	1.90 b	0.67	3.66
TH	1.97	0.16 a	2.18 a	0.56	3.30
T-C	1.44 b	0.04 b	1.27 c	0.52 ab	3.26 ab
T-TH	1.41 b	0.03 b	1.47 c	0.37 b	2.66 b
EDI-C	1.75 ab	0.00 b	1.82 bc	0.66 ab	3.66 ab
EDI-TH	1.99 ab	0.08 b	2.40 ab	0.53 ab	3.40 ab
LDI-C	2.03 ab	0.13 b	2.60 a	0.82 a	4.06 a
LDI-TH	2.50 a	0.38 a	2.65 a	0.79 a	3.84 a
Significance					
RDI	**	***	***	**	*
CL	ns	*	*	ns	ns
RDI*CL	ns	*	ns	ns	ns
Treatment	**	***	***	**	ns
2010					
T	1.54	0.41	3.25	0.41	3.46 ab
EDI	1.61	0.49	3.19	0.44	3.26 b
LDI	1.86	0.57	3.46	0.37	4.14 a
C	1.49	0.49	2.57 b	0.38	3.11 b
TH	1.85	0.49	4.03 a	0.43	4.14 a
T-C	1.34	0.40	2.47 b	0.39	3.46 b
T-TH	1.74	0.42	4.03 a	0.42	3.47 ab
EDI-C	1.36	0.53	2.53 b	0.36	2.53 b
EDI-TH	1.86	0.45	3.84 a	0.52	4.00 ab
LDI-C	1.77	0.55	2.71 b	0.40	3.34 b
LDI-TH	1.96	0.59	4.22 a	0.34	4.95 a
Significance					
RDI	ns	ns	ns	ns	*
CL	ns	ns	***	ns	**
RDI*CL	ns	ns	ns	ns	*
Treatment	ns	ns	*	ns	**

CA, (+)-catechin; EC, (−)-epicatechin; B1, procyanidin B1; B2, procyanidin B2; B3, procyanidin B3. RDI, regulated deficit irrigation, T, rainfed; EDI and LDI, regulated deficit irrigation; CL, crop load; C, control; TH, cluster thinning. Within each column, different letters indicate significant difference among treatments after Tukey test at *p* < 0.05. *, **, *** and ns indicate significant differences at *p* < 0.05, *p* < 0.01, *p* < 0.001 and no significance, respectively.

**Table 4 plants-11-01393-t004:** Effects of regulated deficit irrigation and cluster thinning on grape skin hydroxycinnamic acid, its derivates and stilbenes concentration (mg/kg berry fresh weight) in 2009 and 2010.

	CF	COU	FE	CF-T	COU-T	FE-T	Trans-R
2009							
T	1.2 ab	3.0 a	0.7 a	2.8 a	7.9 b	0.7 ab	0.08 b
EDI	1.3 a	2.6 ab	0.4 b	3.1 a	12.7 a	0.9 a	0.18 a
LDI	0.9 b	2.2 b	0.5 ab	1.9 b	7.2 b	0.6 b	0.10 b
C	1.1	2.6	0.5	2.3 b	10.7 a	0.8	0.11
TH	1.2	2.6	0.6	2.9 a	7.8 b	0.6	0.13
T-C	1.1	3.0	0.8 a	2.6 a	8.1 b	0.7 ab	0.10 b
T-TH	1.2	3.0	0.7 ab	3.0 a	7.8 b	0.6 ab	0.07 b
EDI-C	1.2	2.8	0.3 b	3.0 a	16.6 a	1.0 a	0.13 b
EDI-TH	1.3	2.5	0.5 ab	3.2 a	8.8 b	0.7 ab	0.22 a
LDI-C	0.9	1.9	0.4 ab	1.4 b	7.5 b	0.6 b	0.10 b
LDI-TH	1.0	2.5	0.5 ab	2.5 a	7.0 b	0.6 b	0.10 b
Significance							
RDI	*	*	*	***	***	**	***
CL	ns	ns	ns	**	***	ns	ns
RDI*CL	ns	ns	ns	ns	***	ns	**
Treatment	ns	ns	*	**	***	*	***
2010							
T	0.8 b	1.9 b	0.5 b	2.2 b	4.7 b	0.2	0.06 b
EDI	1.6 a	6.0 a	1.5 a	3.0 a	9.2 a	0.2	0.11 a
LDI	0.7 b	6.0 a	1.2 ab	1.7 b	2.8 c	0.2	0.07 b
C	0.6 b	4.4	1.1	1.9 b	3.4 b	0.2	0.09
TH	1.4 a	4.9	1.0	2.9 a	7.7 a	0.2	0.07
T-C	0.5 d	1.4 b	0.4 b	1.6 c	3.4 bc	0.2	0.05
T-TH	1.2 b	2.5 b	0.6 b	2.9 ab	5.9 b	0.3	0.07
EDI-C	1.0 bc	3.3 b	1.1 b	2.4 bc	4.9 b	0.3	0.11
EDI-TH	2.3 a	8.7 a	1.9 a	3.7 a	13.5 a	0.2	0.10
LDI-C	0.5 cd	8.5 a	1.8 a	1.7 c	2.0 c	0.2	0.09
LDI-TH	0.8 bcd	3.5 b	0.7 b	2.1 bc	3.5 bc	0.2	0.05
Significance							
RDI	***	***	*	**	***	ns	*
CL	***	ns	ns	***	***	ns	ns
RDI*CL	**	***	ns	ns	***	ns	ns
Treatment	***	***	*	***	***	ns	*

Hydroxycinnamic acids tartaric derivatives: CF-T, caftartic acid; COU-T, coutaric acid; FE-T, fertaric acid. Hydroxycinnamic acids: CF, caffeic acid; COU, cumaric acid; FE, feluric acid; Trans-R, trans-resveratrol. RDI, regulated deficit irrigation, T, rainfed; EDI and LDI, regulated deficit irrigation. CL, crop load; C, control; TH, cluster thinning. Within each column, different letters indicate significant difference among treatments after Tukey test at *p* < 0.05. *, **, *** and ns indicate significant differences at *p* < 0.05, *p* < 0.01, *p* < 0.001 and no significance, respectively.

**Table 5 plants-11-01393-t005:** Meteorological and irrigation data for the different irrigation treatments and years of study.

Meteorological Data		2009	2010
Mean Temperature (°C)	Pre-veraison ^1^	19	19
	Post-veraison ^2^	25	26
	Growing season ^3^	22	23
Rainfall (mm)	Pre-veraison	85	137
	Post-veraison	6	5
	Growing season	91	142
	Annual ^4^	402	734
Irrigation (mm)			
EDI	Total	332	207
LDI	Total	274	129

EDI and LDI: regulated deficit irrigation in pre- and post-veraison, respectively. ^1^ Pre-veraison, period from budbreak to veraison; ^2^ Post-veraison, period from veraison to harvest; ^3^ Growing season, period from budbreak to harvest. ^4^ Annual, period from harvest to harvest. Budbreak occurred on 20 March and 26 March in 2009 and 2010, respectively. Veraison started on 15 July and 19 July in 2009 and 2010, respectively.

## Data Availability

Not applicable.

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
