# Peer review of "Improving the Phenolic Content of Tempranillo Grapes by Sustainable Strategies in the Vineyard"

_plants, 2022, doi:10.3390/plants11111393_

Round 1

Reviewer 1 Report

In the present work Esperanza Valdés et al. have proposed some sustainable strategies to improve the phenolic content of Tempranillo grapes vineyard. The work is technical sound and the authors utilized appropriate techniques of analysis. The essential problems of this work is lack of novelty and originality.

Some others criticisms are: 

- The abstract is not linear. There are some sentences that seem to state the exactly contrary of the above sentence like the work performed is full of exceptions but with only two years of results is very difficult to support these data.

-The conclusion section must be improved to better explain the obtained results and their potentiality

Author Response

Improving the phenolic content of Tempranillo grapes by sustainable strategies in the vineyard by Valdés, Talaverano, Moreno, Uriarte, Mancha and Vilanova

Reviewer 1

In the present work Esperanza Valdés et al. have proposed some sustainable strategies to improve the phenolic content of Tempranillo grapes vineyard. The work is technical sound and the authors utilized appropriate techniques of analysis. The essential problems of this work is lack of novelty and originality.

Some others criticisms are:

- The abstract is not linear. There are some sentences that seem to state the exactly contrary of the above sentence like the work performed is full of exceptions but with only two years of results is very difficult to support these data.

The conclusion section must be improved to better explain the obtained results and their potentiality

Thank you for your comments.

We have made the changes in the manuscript (in blue color) as the reviewer recommended.

Reviewer 2 Report

The work is very interesting and from our point of view it can be published as long as the following aspects are improved and revised:
1. It would be convenient to explain in more detail the control techniques carried out in this work (lines 69 to 71) and the specific objectives (lines 80 to 84).
2. Figure 1a and 1b, should improve above all the violet scale symbols that appear within the squares T-C; T-TH; EDI-C; EDI-TH; ILD-C; LDI-TH and when comparing the results of the years 2009 and 2010 (lines 120 to 134), try to specify the meaning of the acronyms in letters and not speak in acronym language.
3. Try to explain in more detail and using as little acronym language as possible, the meaning of Tables 1 to 4.

Other minor aspects and typographical errors that should be checked in the lines:
18. missing close parentheses at RDI
89. specify the meaning of Tr
92. explain at the beginning of the results, the meaning of RDI * CL, without having to refer to the figure or the meaning of the acronyms
210. specify the meaning of TSS

Author Response

Reviewer 2

The work is very interesting and from our point of view it can be published as long as the following aspects are improved and revised:

Thank you for your comments. We have made the changes in the manuscript (in blue color) as the reviewer recommended.
1. It would be convenient to explain in more detail the control techniques carried out in this work (lines 69 to 71) and the specific objectives (lines 80 to 84).
It was done 

2. Figure 1a and 1b, should improve above all the violet scale symbols that appear within the squares T-C; T-TH; EDI-C; EDI-TH; ILD-C; LDI-TH and when comparing the results of the years 2009 and 2010 (lines 120 to 134), try to specify the meaning of the acronyms in letters and not speak in acronym language.
We have made the changes in the Figures. The color has been changed. 

3. Try to explain in more detail and using as little acronym language as possible, the meaning of Tables 1 to 4.
It was done

Other minor aspects and typographical errors that should be checked in the lines:
18. missing close parentheses at RDI
It was done

89. specify the meaning of Tr
It was done. Tr was changed by TR (treatment)

92. explain at the beginning of the results, the meaning of RDI * CL, without having to refer to the figure or the meaning of the acronyms
It was done 

202. specify the meaning of TSS
We change TSS by total solid soluble

Reviewer 3 Report

The authors presented an interesting study using sustainable strategies to improve the quality of Tempranillo grapes, by applying deficit irrigation and crop load level. Despite being interesting and well planned, the paper itself needs corrections and clarifications before its acceptance by Plants. First of all, I suggest that the authors check the English, since some sentences are not clear and minor English mistakes can be found along the manuscript. Regarding the study itself, the results need to be re-organized, presenting first the qualitative analysis of polyphenols and just after the general content of the samples submitted to the different treatments. About the treatments, I suggest that the authors clarify them in their M&M. Some other parts of the text also need some detailing and clarification. The discussion of the total content of each class of phenolics needs special attention, as well as the conclusion and the figures (that should be presented different panels and not different figures, with one caption). The authors should also think about how meaningful is to present the PCA results.  Moreover, maybe the biggest concern is about the biomass/yield results. The authors did not present the analysis of these parameters, or how the treatments changed them. In some parts of the discussion they present some results, but their analysis was not presented as a content of the paper itself. Please, find attached the PDF file with all my detailed comments and corrections. Based on them, I recommend minor review for the paper. 

Author Response

Reviewer 3
The authors presented an interesting study using sustainable strategies to improve the quality of Tempranillo grapes, by applying deficit irrigation and crop load level. Despite being interesting and well planned, the paper itself needs corrections and clarifications before its acceptance by Plants. 
First of all, I suggest that the authors check the English, since some sentences are not clear and minor English mistakes can be found along the manuscript. 

Thank you fore your comments. We have made the changes in the manuscript (in blue color) as the reviewer recommended.
English was revised as the reviewer recommended

Regarding the study itself, the results need to be re-organized, presenting first the qualitative analysis of polyphenols and just after the general content of the samples submitted to the different treatments.
We have decided this organization because is easier to understand the results. First we give a global vision to know the effects of treatments applied and then we perform a detailed analysis of these differences in basis to individual compounds.

 About the treatments, I suggest that the authors clarify them in their M&M. Some other parts of the text also need some detailing and clarification. 
It was done

The discussion of the total content of each class of phenolics needs special attention, as well as the conclusion and the figures (that should be presented different panels and not different figures, with one caption). 
It was done. More discussion was included

The authors should also think about how meaningful is to present the PCA results.
The PCA was done to discriminate among treatments in basis to the polyphenolic family’s concentration. We think theta the PCA permit graphically visualizes the results.

Moreover, maybe the biggest concern is about the biomass/yield results. The authors did not present the analysis of these parameters, or how the treatments changed them. In some parts of the discussion they present some results, but their analysis was not presented as a content of the paper itself. 
Leaf area analysis is now included in Material and Methods section. Data of leaf area is included in the text because is few information to include another table in the manuscript. We have included a new table with climatic conditions and irrigation data by year of study.

Please, find attached the PDF file with all my detailed comments and corrections. Based on them, I recommend minor review for the paper.
Some clarifications about reviewer comments:
In phenolic composition, the compounds are classified in families or groups but no classes, for this reason we use families in the text.
Line 24: We use trend or tendency when significant differences were not found.
Line 29: The year was considered independently compared. 
Line 109: We can ensure “In our study, dry and warm meteorological conditions during pre-veraison of 2009, caused higher water stress level vines” because is a result of previous study performed by our group. Now we have included the reference [36] in the text. The same situation for lines 114.
Line 123: We are giving the leaf area data in the sentence (not in table) as follow because in low information to include in a table. 
We have added the method of leaf area in materials and method section.
All comments in the .pdf were revised as the reviewer recommended (in blue)

Lines 452: “Quantification of non-commercial compounds was made using the straight calibration compound belonging to the same family next in the order of elution.family next in the order of elution”. Replace “family” by “class”.  Not only the RT indicates a class of a compound, but specially the UV spectra. Please include this.
This sentence refers to quantification, not identification. As previously indicated, the use of UV-spectra for the identification of phenolic compounds has been included.
